# Evaluating Farmers' Access to Agricultural Information: Evidence from Semi-Arid Region of Rajasthan State, India

**Ishwar S. Parmar** [1,*] , **Peeyush Soni** [2] , **John K. M. Kuwornu** [3] **and Krishna R. Salin** [3]

1 Production (SAP), Al Rawabi Dairy Co. LLC, Al Khawaneej, 50368 Dubai, UAE
2 Department of Agricultural & Food Engineering, Indian Institute of Technology, Kharagpur 721302, India; soni@iitkgp.ac.in
3 Department of Food, Agriculture and Bioresources, School of Environment, Resources and Development, Asian Institute of Technology, Pathum Thani 12120, Thailand; jkuwornu@ait.asia (J.K.M.K.); salinkr@ait.asia (K.R.S.)
* Correspondence: parmaris@hotmail.co.uk; Tel.: +971-505249396

**Abstract:** The rural farmers in western Rajasthan State are uneducated and most of the applications of Information and Communication Technology (ICT) are demonstrated and run in the English Language. The majority of these rural farmers who are illiterates with a very low level of understanding of the English Language find it difficult to take advantage of the availability of ICT to facilitate their access to information for their farm businesses. This study examined the role of ICT in enhancing the farmers' access to production and marketing information in western Rajasthan State in India. Primary data was collected from 133 farmers consisting of 71 ICT users and 62 Non-ICT users through questionnaire administration. The results of the Analysis of Variance test regarding the farmers' access to different types of production and marketing information revealed that the user type (i.e., ICT versus Non-ICT user) significantly explains the differences in farmers' access to the different types of marketing and production information. These results are consistent with the empirical results of the student's *t*-test that farmers' access to different types of production and marketing information from ICT sources is significantly higher than from Non-ICT sources. Consistently, the empirical results of the multiple regressions revealed that the percentage of production and marketing information obtained from ICT sources had positive significant influence on the farmers' access to marketing and production information; and that the percentage of marketing and production information obtained from Non-ICT sources had negative significant influence on the farmers' access to marketing and production information. These results suggest that ICT sources of marketing and production information play a crucial role in the farmers' access to this information for their business operations. The implication is that proper education and training of farmers (especially the female farmers) about the utilization of ICT sources to accelerate access to information is crucial.

**Keywords:** semi-arid region; adoption of ICT; agricultural information; farmers; education; India

## 1. Introduction

Agriculture is the crucial sector of the Indian economy, predominantly because the majority (64.2%) of the rural population of India is dependent on it [1]. Indian agriculture accounts for 18% of the country's GDP, and approximately 62.5% of the Indians derive their livelihood from the horticultural sector [2]. Farming in the western parched region of the Rajasthan State is principally rain fed. Farmers in this region of the country are poor and have limited access to market information, innovation, and strategies, leading to low farm productivity [3].

Information and Communication Technology (ICT) is critical to the dissemination of market information to farmers. ICT also facilitates the dissemination of information on business skills and production practices to the farmers [4]. In India, the provision of agricultural knowledge to farmers is administered by the government agencies (i.e., the government Agricultural Extension System).

The information exchange between extension and the farmers is presented in Figure 1.

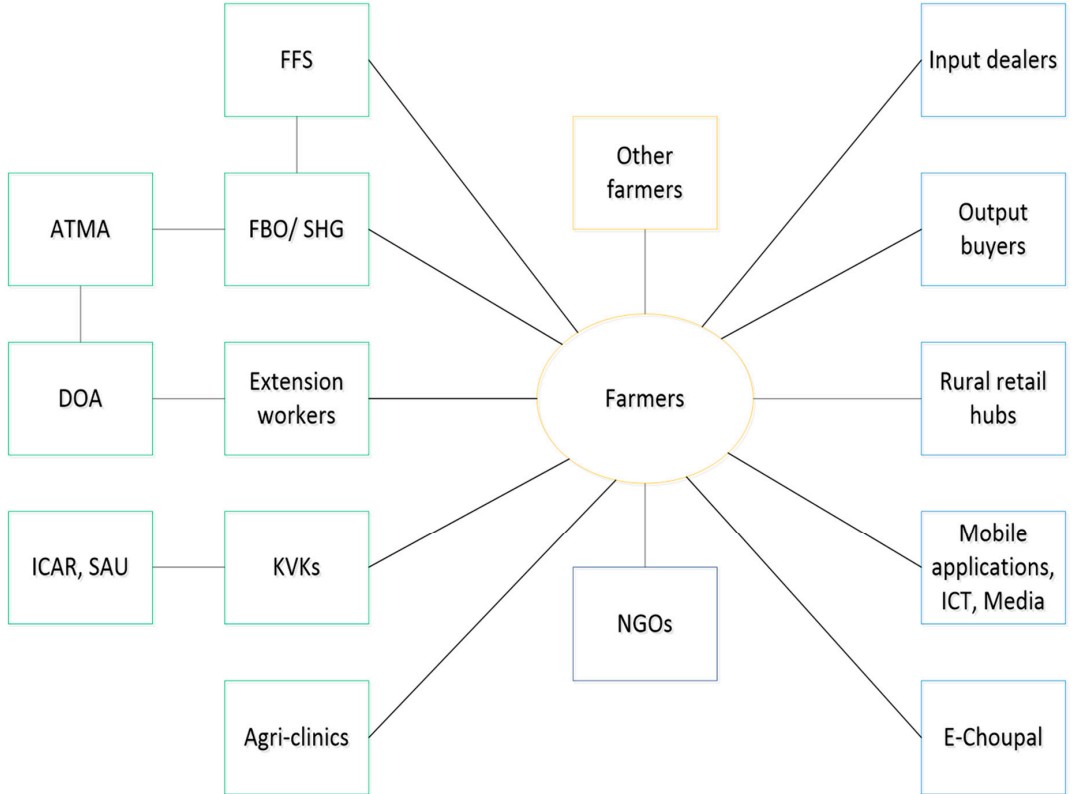

**Figure 1.** Information exchange between extension and farmers in India. Source: Glendenning et al., 2010.

Notes: Information flow is shown by the line linking the boxes. The green boxes refer to the public sector, and the blue ones to the Private sector. ATMA denotes Agricultural Technology Management Agency, DoA denotes Department of Agriculture, ICAR denotes Indian Council for Agricultural Research, FFS denotes farmer field school, FBO/SHG denotes farmer-based organization/self-help group, SAU denotes state agricultural university, KVK denotes Krishi Vigyan Kendra (farm science center), and NGO denotes non-governmental organization.

Information can be defined as data that is systematically collected and relevant, and serves as a resource [5,6]. Agricultural Information is the systematically collected, published and unpublished data relating to the agricultural sector [7]. The users of agricultural information include researchers, organizers, policymakers, instructors, students, field laborers, program administrators, and farmers [8]. In the contemporary framework, the farming community can be categorized as literate and illiterate farmers based on their educational levels. In the Rajasthan State in India, agricultural information is provided to literate farmers through Krishi Vigyan Kendra (Farm Science Center) in the form of vocational learning or electronic media, but this approach to information provision could not be accessed by illiterate farmers. Traditionally, illiterate farmers can be instructed by audio-visual modes.

The challenges faced by the agriculture sector in Rajasthan are increasing gap between demand and availability of water, scanty and uncertain rainfall, deteriorating quality of land and underground water, large gap between potential and realized yield of crops and high inter-year variation in productivity [9]. The vulnerability of the farmers increases with changing environmental and socio-economic conditions [10].

The economic rationale for the farmers' access to information is to enable them to the manage risks and uncertainties regarding production and marketing of their produce. The better the farmers manage these risks and uncertainties the more profitable their businesses become. ICT facilitates awareness and access to market information among the farmers [4]. There are more than 200 ICT development agencies in different stages of implementation in India e.g., Bhoomi, Drishtee. These agencies provide information relating to, for instance, climate reports, and marketing information e.g., Krishi Vigyan Kendras/Farm Science Centers at Ahmednagar, Baramati) [11].

Farmers obtain production and marketing information from various sources [12–14]. Some of these sources utilize ICTs while the others are Non-ICT sources. The effect of age, education and farm characteristics on adoption of ICT has been extensively documented in the literature [15–20]. Consequently, this study examined the effect of percentage of information sought from ICT sources and the percentage of information sought from Non-ICT sources on the farmers' overall access to production and marketing information. The socio-economic and some farm–level characteristics have also been included in the analysis in this study. It is worth noting that the findings regarding the effect of age on the adoption of ICT are contradictory. Thus, whereas some studies revealed a positive influence of age on the adoption of ICT, other studies revealed negative influence [15–19].

In the western infertile part of Rajasthan, the farmers' access to information is foreseen to be extremely poor, ill-timed, less credible, and not cost-effective [3]. Acceptance of mobile phones as an advent of delivering agriculture-related data relies on the accessibility of mobile network in the rural terrains.

Information technology is very important for forecasting the climate. Farmers can obtain climate and weather updates through communication satellites and different technologies. They can also obtain information on market prices of the products. Correspondingly, the appropriate agricultural information may help to minimize the prices paid for agricultural inputs by the farmers, increase the quality of the produce, and increase the marketing prices and profits [21].

The objectives of this study are twofold as follows: (i) to compare access to production and marketing information between two categories of farmers (i.e., farmers who sought information from ICT sources and those that sought information from Non-ICT sources; (ii) to empirically examine the effects of the percentage of information sought from ICT sources, and the percentage of information sought from Non-ICT sources on famers' overall access to production and marketing information.

*Conceptual Framework*

Farmers require production and marketing and information to support their business operations. These farmers use the information obtained from a source or combination of sources for their activities. They utilize ICT and Non-ICT sources to obtain agricultural information for the farm activities.

The conceptual framework for this study hinges on the differences between farmers' access to production and marketing information based on the sources of the information (i.e., ICT sources and Non-ICT sources), (Figure 2).

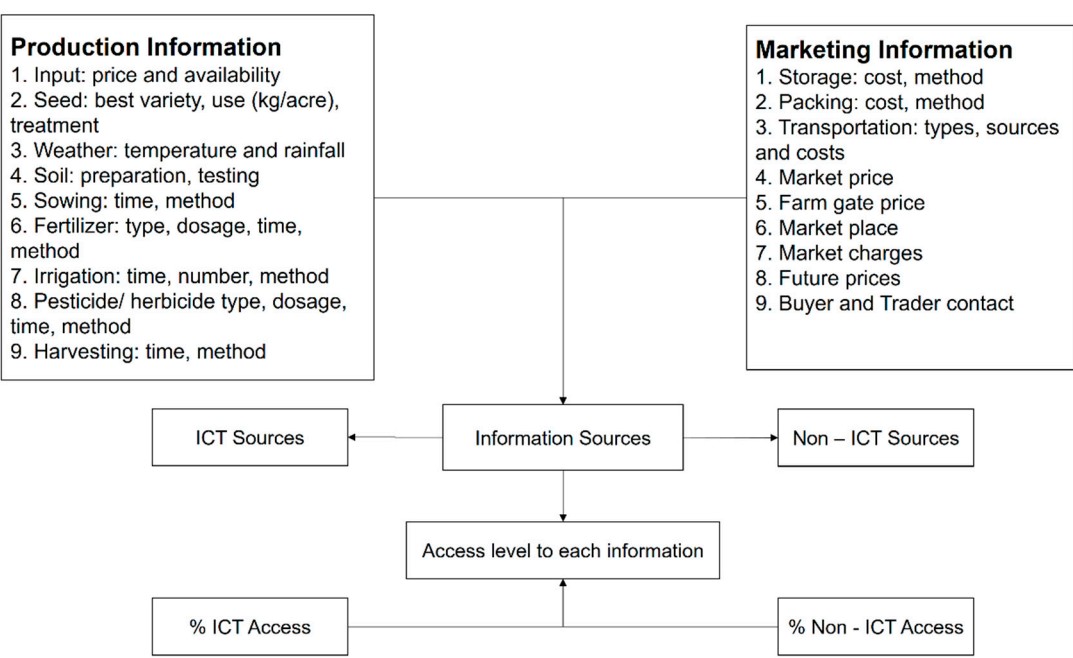

**Figure 2.** Conceptual framework.

## 2. Materials and Methods

### 2.1. Area of Study

Rajasthan is the largest state of India having land area of approximately 342,239 km$^2$. It has 33 districts and Jaipur is its capital city. It is situated on the western side of the nation. The number of inhabitants in Rajasthan is 68.54 million [22].

Agricultural production in Rajasthan State is very tough due to the harsh dry climate found in the major parts of the State [23]. However, agriculture employs 64.2% of the rural working population of the State [24]. Agriculture, including animal husbandry, contributed 24.59% to the State's Gross Domestic Product (GDP) during 2012–2013 [25]. The growth of the agriculture sector, therefore, has an important influence on the lives of people dependent on agriculture. The high level of illiteracy (61.4%) among the rural community of Rajasthan is the foremost social reasons for agriculture vulnerability [24]. Agricultural production in Rajasthan is confronted with land scarcity not only due to the unfavourable topography but also competition for land by the industrial sector. The heavy dependence of agriculture in Rajasthan on the monsoon rainfall makes it more vulnerable in the context of the changing climate [26,27]. The rainfall pattern varies for different regions of Rajasthan. Similarly, the variations in climate are not uniform across the State and, therefore, single contingency plan for agriculture sustainability cannot be formulated for the whole State and a more detailed regional plan is needed [28].

Rajasthan State has four major physiographic regions as follows: the western desert (Thar Desert), the Aravalli hills, the eastern plains and the south-eastern plateau (Hadoti Plateau). About 62% of the area consists of sandy plains, and consequently known as the Desert State of India. The Aravalli hills running diagonally across the State form the geomorphic and climatic boundary of the desert in the east. The western part merges into the Pakistan desert. The Aravalli Range is the major water divide in the State. The area in the east is well drained by several integrated drainage systems, whereas that in the west has only one, the Luni drainage system. The climate is characterized by low rainfall with erratic distribution, extremes of diurnal and annual temperatures, low humidity and high wind velocity. The arid climate has marked variations in diurnal and seasonal ranges of temperature, characteristic of warm-dry continental climates. During summer (March to June), the maximum temperature generally varies between 40 °C and 49 °C. Night temperatures decrease considerably, to 20–29 °C. January

is the coldest month. During winter (December to February), minimum temperatures may fall to −2 °C at night. Occasional secondary Western disturbances, which cross mostly western, northern and eastern Rajasthan during the winter months cause light rainfall and increased wind speeds which result in a wind-chill effect. The average annual rainfall ranges from less than 100 mm to 400 mm. The State is divided into 10 agro-climatic zones as follows: Arid western plain, Irrigated north western plain, Hyper arid partial irrigated zone, Internal drainage dry zone, Transitional plain of Luni basin, Semi-arid eastern plains, Flood prone eastern plain, Sub-humid southern plains, Humid southern plains, and Humid south eastern plain [29].

The map of Rajasthan State and the studied districts within the State (i.e., Jalore, Pali and Sirohi) are shown by Figure 3, and the divisions and districts of the State are presented in Table 1.

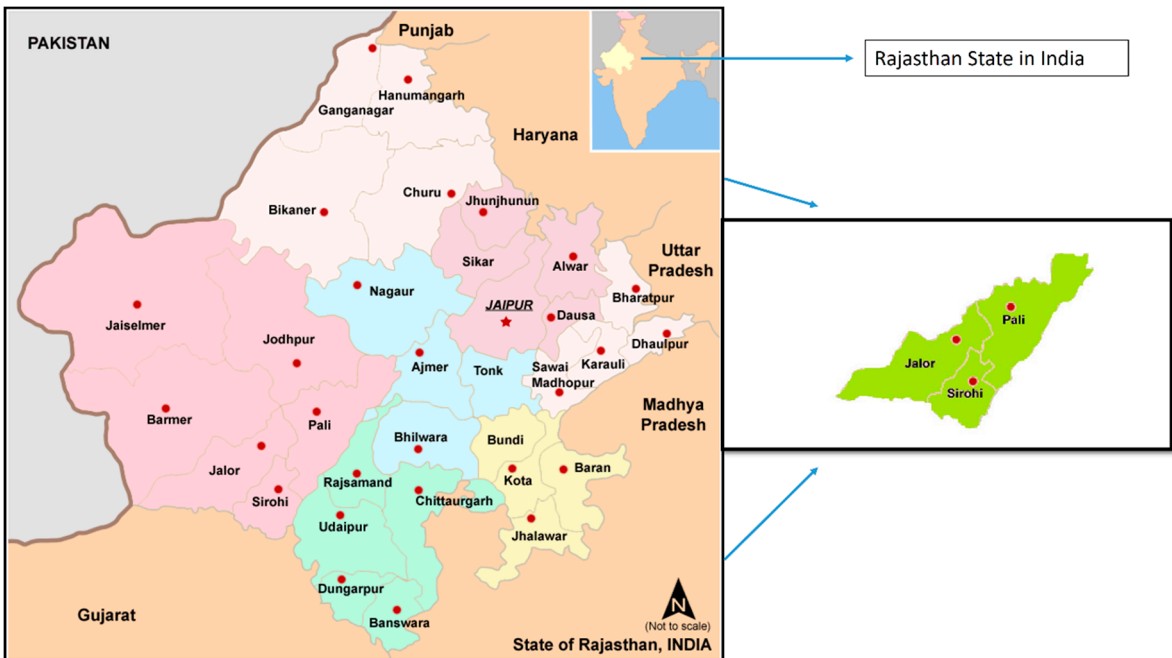

**Figure 3.** Map of Rajasthan State in India showing Jalore, Pali and Sirohi districts. Source: https://commons. wikimedia.org/wiki/File:Map_rajasthan_dist_7_div.png http://www.mapsofindia.com/maps/rajasthan.

**Table 1.** Divisions and districts of Rajasthan state, India.

| Divisions | Districts |
| --- | --- |
| Jaipur | Jaipur, Alwar, Jhunjhunu, Sikar & Dausa |
| Jodhpur | Barmer, Jaisalmer, Jalore, Jodhpur, Pali & Sirohi |
| Ajmer | Ajmer, Bhilwara, Nagaur & Tonk |
| Udaipur | Udaipur, Banswara, Chittorgarh, Pratapgarh, Dungarpur & Rajsamand |
| Bikaner | Bikaner, Churu, Sri Ganganagar & Hanumangarh |
| Kota | Baran, Bundi, Jhalawar & Kota |
| Bharatpur | Bharatpur, Dholpur, Karauli & Sawai Madhopur |

Source: Districts of Rajasthan state, India (https://www.mapsofindia.com/maps/rajasthan/districts/).

## 2.2. Sampling and Data

The current study was carried out in three semi-arid districts (i.e., Jalore, Pali and Sirohi) of the Rajasthan state, India, in May 2016 to July 2016 to evaluate farmers' access to agricultural information from ICT and Non-ICT sources, and to examine the factors influencing access to the information.

A multistage sampling technique was used to select the households for data collection. First, the purposive sampling technique was employed to select the Rajasthan State and three districts in the State (i.e., Jalore, Pali and Sirohi) due to high level of agricultural production activities. Second,

simple random sampling was used to select the farmers in the districts for interview and data collection. The questionnaire for data collection was initially developed and pretested to remove all ambiguities before finalising for the actual data collection. By this process, we ensured that quality data is obtained for the analysis. The specific questions in the questionnaire relating to access to production and marketing information were based on a five-point likert scale as follows: very low (1), low (2), medium (3), high (4), and very high (5). The population of farmers in the three districts was 689,960 comprising 200,091 in Pali, 387,143 in Jalore, and 102,726 in Sirohi [24]. In all, data from 133 farmers was collected for the study: Pali (51), Jalore (15), and Sirohi (67). Out of 133 respondent farmers interviewed for the data collection, 102 were both livestock breeders and crop farmers, and 31 farmers were crop farmers only. Also, the 133 sampled farmers consisted of 71 ICT users and 62 Non-ICT users. The farmers who were interviewed for the data collection were household heads. These farmers were the main source of income and take major decisions for their households.

### 2.3. Methods of Analysis

Descriptive statistics were used to present the socio-economic characteristics of the farmers. Weighted average index was used to examine the farmers' access to production and marketing information. The One-way Analysis of Variance test (ANOVA) was used to examine significant differences in access to different types of production and marketing information. Furthermore, the student's t-test was used to examine the differences in access to different types of production and marketing information between the users of ICT and Non-users of ICT. Finally, multiple regression was employed to examine the factors influencing farmers' access to production and marketing information.

### 2.3.1. Description of Variables

The sampled rural farmers accessed the information from 2 types of sources as follows: 8 ICT sources: Television, Radio, Mobile phone, Landline phone, Community loudspeaker, Computer, Internet, and Newspaper; and 8 Non-ICT sources: Farm science center, Public Extension Agent, Input merchants, Output merchants, Private advisor, other farmers, Relatives and Friends, and Others.

The percentage of information accessed from ICT and Non-ICT sources are specified in Equations (1) and (2) as follows:

$$\text{Percentage ICT Access} = \left[ \left( \sum S_i F_i / SF \right) * 100 \right] / 2 \tag{1}$$

where $S_i$ = ICT source $i$; $F_i$ = Frequency of ICT source $i$. The frequency of farmers' access to information from the individual ICT sources were assigned values as follows: None (0), Yearly (1), Seasonal (2), Monthly (3), Fortnight (4), Weekly (5), Daily (6). $S$ = Total ICT sources; $F$ = Maximum frequency = 6 (daily).

$$\text{Percentage Non} - \text{ICT Access} = \left[ \left( \sum S_j F_j / SF \right) * 100 \right] / 2 \tag{2}$$

where $S_j$ = Non-ICT source $j$; $F_j$ = Frequency of Non-ICT source $j$. The frequency of farmers' access to information from the individual Non-ICT sources were assigned values as follows: None (0), Yearly (1), Seasonal (2), Monthly (3), Fortnight (4), Weekly (5), Daily (6). $S$ = Total Non-ICT sources; $F$ = Maximum frequency = 6 (daily). We have used equal number of ICT and Non-ICT Sources. The % ICT source from which the information was accessed is calculated based on Total ICT sources. Similarly, the % Non-ICT source from which the information was accessed is calculated based on Total Non-ICT sources. Therefore, to calculate the percentage of ICT sources of information accessed out of Total ICT + Non-ICT accessed, it was divided by 2 as specified in Equations (1). This applies to the percentage of Non-ICT sources of information accessed out of Total ICT + Non-ICT accessed as specified in Equation (2).

### 2.3.2. Weighted Average Index

The weighted average index (WAI) used to examine the famers' access to production and marketing information is specified in Equation (3) based on the values of the five point Likert scale as follows:

$$WAI = [f_{VH}(5) + f_H(4) + f_M(3) + f_L(2) + f_{VL}(1)]/N \qquad (3)$$

WAI = Weighted Average Index; $f_{VH}$ = Frequency of very high access; $f_H$ = Frequency of high access; $f_M$ = Frequency of moderate access; $f_L$ = Frequency of low of access; $f_{VL}$ = Frequency of very low access; and N = Total number of farmers. WAI was calculated for each type of production and marketing information. Thereafter, the averages of all types of production information were used to derive the production mean for the first multiple regression analysis. Similarly, the averages of all types of marketing information were used to derive the marketing information mean for second multiple regression analysis.

### 2.3.3. Regression Analysis

A multiple regression model was employed to examine the factors influencing the level of access to information as follows:

$$Y = \alpha_i + \beta_i X_i + U_i \qquad (4)$$

where $Y$ denotes the farmers' level of access to information, $\alpha_i$ and $\beta_i$ are the coefficients to be estimated, $X_i$ denotes explanatory variables (i.e., the percentage of information accessed from ICT sources, the percentage of information accessed from Non-ICT sources, and the socio-economic and farm–level factors, and $U_i$ is the error term.

### 2.4. Socio-Economic Characteristics of the Farmers

Approximately 66.9% of our respondents are crop producers (i.e., wheat, maize, pulses and sesame) and livestock breeders. The key livestock were cattle and buffalo [30].

Seventy one of our respondents who solicited information from ICT sources were males and no female solicited information from ICT sources. Of the respondents who solicited information from Non-ICT sources 55 were males and 7 were females [31,32]. Regarding marital status, 70 married farmers solicited information from ICT sources and 55 married farmers solicited information from Non-ICT sources. Similar observations were documented by [32,33], (Table 2).

Sixty nine of our respondent farmers who obtained at least secondary school education obtained information from ICT sources, and 54 of the farmers consisting of illiterates and those who obtained primary school education solicited information from Non-ICT sources (Table 2). This is consistent with the observation by [30].

Fifty four of the sampled farmers who solicited information from ICT sources were in the age group 34–59 years, and 43 in the same age group solicited information from Non-ICT sources. This observation is similar to [34–37]. Forty four farmers who had farming experience of 21–39 years, and 7 farmers had farming experience of 40–58 years solicited information from ICT sources. Thirty three farmers who had farming experience of 21–39 years, and 18 farmers had farming experience of 40–58 years solicited information from Non-ICT sources (Table 2). Sixty seven of the farmers who solicited information from ICT sources had at most 25 acres of landholding whereas 4 farmers who solicited information from ICT sources had landholding size in the range 25–50 acres. Similarly, 55 of the farmers who solicited information from Non-ICT sources had at most 25 acres of landholding whereas 6 farmers who solicited information from Non-ICT sources had landholding size in the range 25–50 acres [30].

**Table 2.** Socio-economic characteristics of the farmers.

| Variable | ICT Users | | | | Non-ICT Users | | | |
|---|---|---|---|---|---|---|---|---|
| | Frequency | Mean | Standard Deviation | Standard Error | Frequency | Mean | Standard Deviation | Standard Error |
| **Gender** | | | | | | | | |
| Male | 71 | | | | 55 | | | |
| Female | 0 | | | | 7 | | | |
| **Marital Status** | | | | | | | | |
| Single | 1 | | | | 0 | | | |
| Married | 70 | | | | 55 | | | |
| Divorced | 0 | | | | 0 | | | |
| Separated | 0 | | | | 0 | | | |
| Widow | 0 | | | | 7 | | | |
| **Education Level** | | | | | | | | |
| Illiterate | 1 | | | | 25 | | | |
| Primary | 1 | | | | 29 | | | |
| Secondary | 33 | | | | 6 | | | |
| Senior Secondary | 14 | | | | 1 | | | |
| Bachelor | 19 | | | | 1 | | | |
| Master or higher | 3 | | | | 0 | | | |
| **Age (Years)** | | 47.44 | 10.04 | 1.193 | | 51.52 | 10.55 | 1.34 |
| 21–33 | 8 | | | | 1 | | | |
| 34–46 | 23 | | | | 23 | | | |
| 47–59 | 31 | | | | 20 | | | |
| 60–72 | 9 | | | | 18 | | | |
| **Total Family Members** | | 5.35 | 1.725 | 0.205 | | 6.61 | 3.13 | 0.398 |
| 2–6 | 58 | | | | 42 | | | |
| 7–11 | 13 | | | | 13 | | | |
| 12–16 | 0 | | | | 7 | | | |
| **Family Labor** | | 2.93 | 1.073 | 0.127 | | 3.81 | 2.14 | 0.272 |
| 1–3 | 49 | | | | 33 | | | |
| 4–6 | 22 | | | | 21 | | | |
| 7–9 | 0 | | | | 8 | | | |
| **Years of Farming Experience** | | 25.48 | 9.648 | 1.145 | | 31.06 | 11.39 | 1.45 |
| 2–20 | 20 | | | | 11 | | | |
| 21–39 | 44 | | | | 33 | | | |
| 40–58 | 7 | | | | 18 | | | |
| **Land-holding Size (acres)** | | 8.721 | 8.17 | 0.97 | | 12.5 | 13.43 | 1.71 |
| 0–25 | 67 | | | | 55 | | | |
| 25–50 | 4 | | | | 6 | | | |
| 50–75 | 0 | | | | 0 | | | |
| 75–100 | 0 | | | | 1 | | | |
| **Farm Income (Rs.)** | | 14215.4 | 9866.8 | 1170.98 | | 20137.1 | 13388.63 | 1700.36 |
| 0–19,999 | 46 | | | | 31 | | | |
| 20,000–39,999 | 24 | | | | 27 | | | |
| 40,000–59,999 | 1 | | | | 3 | | | |
| 60,000–79,999 | 0 | | | | 1 | | | |
| **Off-farm Income (Rs.)** | | 22140.8 | 17865.4 | 2120.23 | | 13177.4 | 14909.04 | 1893.45 |
| 0–19,999 | 36 | | | | 42 | | | |
| 20,000–39,999 | 21 | | | | 15 | | | |
| 40,000–59,999 | 11 | | | | 5 | | | |
| 60,000–79,999 | 3 | | | | 0 | | | |
| **Total Income (Rs.)** | | 36356.3 | 14697.1 | 1744.23 | | 33314.5 | 15866.56 | 2015.06 |
| 0–19,999 | 7 | | | | 11 | | | |
| 20,000–39,999 | 40 | | | | 35 | | | |
| 40,000–59,999 | 18 | | | | 12 | | | |
| 60,000–79,999 | 5 | | | | 3 | | | |
| 80,000–99,999 | 1 | | | | 1 | | | |
| **Employment Type** | | | | | | | | |
| Part time | 57 | | | | 35 | | | |
| Full time | 14 | | | | 27 | | | |

Source: Field survey, 2016.

## 3. Empirical Results and Discussion

### 3.1. Access to Production Information

The descriptive statistics of farmers' access to different types of production information are presented in Table A1 and Figure A1 in the Appendix A. The results regarding differences in access to different types of production information by ICT users and Non-ICT users are presented in Table 3. It is worth noting that the user types under consideration in this study are Information and Communication Technology (ICT) and Non-ICT users. Thus, in this study user type on one hand and ICT and Non-ICT users, on the other hand, are used interchangeably.

**Table 3.** Results of the One-way ANOVA regarding the influence of user type on access to different types of production information.

| Access to Production Information Sought/User Type | *p*-Value |
|---|---|
| 1. Input: price and availability<br>2. Seed: best variety, use (kg/acre), treatment<br>3. Weather: temperature and rainfall<br>4. Soil: preparation, testing<br>5. Sowing: time, method<br>6. Fertilizer: type, dosage, time, method<br>7. Irrigation: time, number, method<br>8. Pesticide/herbicide type, dosage, time, method<br>9. Harvesting: time, method<br>Production Information (Mean) | 0.000 *** |

*** denotes significant at 1% level. The F-statistic = 64.821.

There is significant difference between ICT and Non-ICT users regarding their access to different types of production information ($p < 0.1$).

The Cronbach's alpha ($\alpha$) of 0.978, (i.e., 97.8%) indicates a high level of internal consistency for the level of access to production information and 2.2% error variance in the level of access to production information.

The student's *t*-test was performed to examine in detail the differences in access to different types of production information between ICT users and Non-ICT users (Table 4). The results revealed significant differences between ICT and Non-ICT users ($p < 0.01$). Specifically, the results revealed that ICT users obtained more production information than Non-ICT users as reported in the literature [38–42].

**Table 4.** Results of student's *t*-test of the differences in access to different types of production information.

| Type of Production Information | Mean Difference | *T*-Statistic | *p*-Value |
|---|---|---|---|
| Input: price & availability | 2.550 | 19.256 | 0.000 *** |
| Seed: best variety, use (kg/acre), treatment | 2.538 | 20.559 | 0.000 *** |
| Weather: temperature and rainfall | 2.648 | 20.819 | 0.000 *** |
| Soil: preparation, testing | 2.516 | 20.474 | 0.000 *** |
| Sowing: time, method | 2.615 | 20.965 | 0.000 *** |
| Fertilizer: type, dosage, time, method | 2.603 | 21.408 | 0.000 *** |
| Irrigation: time, number, method | 2.761 | 18.558 | 0.000 *** |
| Pesticide/herbicide type, dosage, time, method | 2.478 | 19.786 | 0.000 *** |
| Harvesting: time, method | 2.673 | 21.396 | 0.000 *** |
| Production Information (Mean) | 2.59733 | 32.688 | 0.000 *** |

*** denotes significance at 1% level.

As a prelude to presenting the results of the factors influencing the overall level of access to production information, Figure 4 shows that the level of access to production information (mean) increased as the equivalent ICT rank increased. The two clusters in the figure indicate high values of access to production information by ICT users and the low values of access to production information by Non-ICT users.

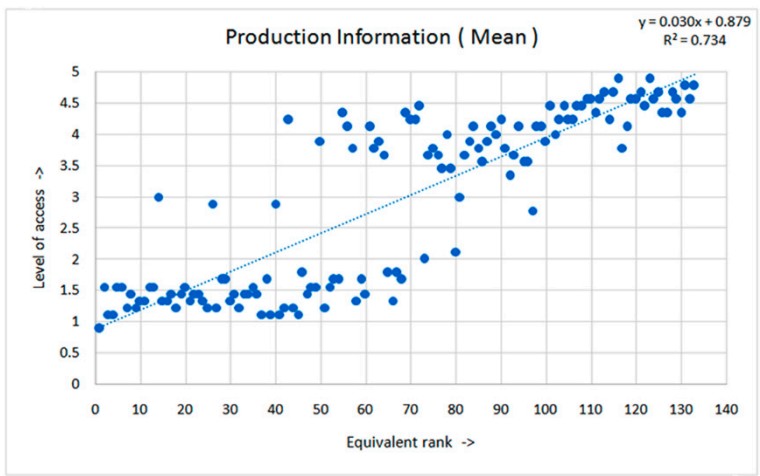

**Figure 4.** Level of access of production information (mean) vs. equivalent ICT rank.

The results of the regression of the factors influencing farmers' access to production information are presented as follows. The adjusted $R^2$ value of 0.811 indicates that approximately 81% of the variations in the level of access to production information are explained by the independent variables (Table 5). The percentage of production information obtained from ICT sources had positive significant influence on the farmers' level of access to production information ($p < 0.01$). On the contrary, the percentage of production information obtained from Non-ICT sources had a negative significant influence on the farmers' level of access to production information ($p < 0.01$), (Table 5). The gender variable is also significant ($p < 0.05$). These results suggest that ICT sources of production information play a crucial role in farmers' access to production information and that the male farmers have more access to production information than the female farmers as documented by previous studies [43–50].

**Table 5.** Multiple regression results of the factors influencing level of access to production information.

| Variables ($X_i$) | Unstandardized Coefficients | | Standardized Coefficients | T-Statistic | p-Value |
|---|---|---|---|---|---|
| | B | Std. Error | Beta | | |
| Constant ($\alpha$) | 0.789 | 0.678 | | 1.165 | 0.246 |
| $X_1$ Percentage ICT Access | 0.152 | 0.013 | 0.853 | 11.952 | 0.000 *** |
| $X_2$ Percentage Non-ICT Access | −0.088 | 0.013 | −0.292 | −7.015 | 0.000 *** |
| $X_3$ Gender | 0.677 | 0.266 | 0.11 | 2.543 | 0.012 ** |
| $X_4$ Educational level | 0.02 | 0.061 | 0.024 | 0.323 | 0.747 |
| $X_5$ Family labor | −0.027 | 0.038 | −0.033 | −0.700 | 0.485 |
| $X_6$ Relative Farming Experience [1] | 0.053 | 0.555 | 0.005 | 0.095 | 0.925 |
| $X_7$ Landholding size | 0.000 | 0.005 | −0.002 | −0.046 | 0.964 |
| $X_8$ Employment type | −0.217 | 0.166 | −0.073 | −1.304 | 0.195 |
| $X_9$ Off-farm income | −0.000005 | 0.000 | −0.059 | −1.016 | 0.312 |
| $R^2$ | | 0.823 | | | |
| Adjusted $R^2$ | | 0.811 | | | |
| F statistic | | 63.756 | | | |
| N | | 133 | | | |

Dependent Variable: Production Information (Mean). *** and ** denote significance at 1% and 5% levels respectively.
[1] Multicollinearity exists between age and the number of years of farming experience as indicated by the values of their variance inflation factor (VIF) of 14.591 and 16.746 respectively; VIF greater than 10 implies multicollinearity. Therefore, we generated a new variable 'relative farming experience' by dividing the number of years of farming experience by the age of the farmer. The VIF of the new variable 'relative farming experience' is 1.679.

### 3.2. Access to Marketing Information

The descriptive statistics of farmers' access to different types of marketing information are presented in Table A2 and Figure A2 in the Appendix A. The results regarding differences in access to different types of marketing information by ICT users and Non-ICT users are presented in Table 6.

**Table 6.** Results of the One-way ANOVA regarding the influence of user type on access to different types of marketing information.

| Access to Marketing Information Sought/User Type | *p*-Value |
|---|---|
| 1. Storage: cost, method<br>2. Packing: cost, method<br>3. Transportation: types, sources and costs<br>4. Market price<br>5. Farm gate price<br>6. Market place<br>7. Market charges<br>8. Future prices<br>9. Buyer and Trader contact<br>Marketing Information (Mean)<br>Information on animal husbandry and fisheries | 0.000 *** |

*** denotes significance at 1% level. The F-statistic = 62.701.

There is a significant difference between ICT and Non-ICT users across their access to different types of marketing information ($p < 0.1$). Indeed, differences in access to marketing information among ICT and Non-ICT users have been documented in the literature [51].

The survey data revealed that the predominant marketing information, i.e., market place, market price, future price and transportation (types, sources and costs) were accessed by the farmers more than the other types of marketing information.

The Cronbach's alpha ($\alpha$) of 0.979, i.e., 97.9%, indicates a high level of internal consistency for the level of access to marketing information and 2.1% error variance in the level of access to marketing information.

The student's *t*-test was performed to examine the differences in access to different types of marketing information between ICT users and Non-ICT users (Table 7). Similar to the results of farmers' access to production information presented previously, the overall results revealed significant differences in farmers' access to marketing information between ICT and Non-ICT users ($p < 0.01$). Thus, ICT users obtained more marketing information than Non-ICT users as found by previous studies [39–42].

**Table 7.** Results of student's *t*-test of the differences in access to different types of marketing information.

| Type of Marketing Information | Mean Difference | *T*-Statistic | *p*-Value |
|---|---|---|---|
| Storage: cost, method | 2.672 | 21.568 | 0.000 *** |
| Packing: cost, method | 2.754 | 21.329 | 0.000 *** |
| Transportation: types, sources and costs | 2.621 | 21.954 | 0.000 *** |
| Market price | 2.657 | 20.059 | 0.000 *** |
| Farm gate price | 2.533 | 20.565 | 0.000 *** |
| Market place | 2.721 | 22.247 | 0.000 *** |
| Market charges | 2.671 | 20.069 | 0.000 *** |
| Future prices | 2.863 | 22.008 | 0.000 *** |
| Buyer & Trader contact | 2.535 | 20.901 | 0.000 *** |
| Marketing Information (Mean) | 2.66955 | 34.636 | 0.000 *** |

*** denotes significance at 1% level.

The level of access to marketing information (mean) increased as equivalent ICT rank increased. The two clusters in Figure 5 indicate high values of access to marketing information by ICT users and the low values of access to marketing information by Non-ICT users (Figure 5). These results are similar to the results on the farmers' access to production information presented previously (Figure 4).

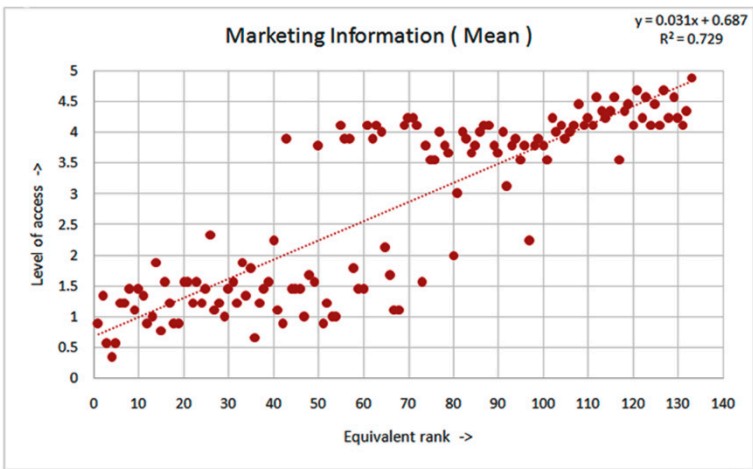

**Figure 5.** Level of access of marketing information (mean) vs. equivalent ICT rank.

The results of the regression of the factors influencing farmers' access to marketing information are presented as follows. The adjusted $R^2$ value of 0.808 indicates that approximately 81% of the variations in the level of access to marketing information are explained by the independent variables (Table 8). Similar to the results regarding the effect of ICT sources on farmers' access to production information, the marketing information obtained from ICT sources had positive significant influence on the farmers' level of access to marketing information ($p < 0.01$). On the contrary, the marketing information obtained from Non-ICT sources had a negative significant influence on the farmers' level of access to marketing information ($p < 0.01$). The gender variable is also significant ($p < 0.05$). These results suggest that ICT sources of marketing information play a crucial role in farmers' access to marketing information and that the male farmers have more access to marketing information than the female farmers as documented by previous studies [43–50].

**Table 8.** Multiple regression results of the factors influencing level of access to marketing information.

| Variables ($X_i$) | Unstandardized Coefficients | | Standardized Coefficients | T-Statistic | p-Value |
|---|---|---|---|---|---|
| | B | Std. Error | Beta | | |
| Constant ($\alpha$) | 0.765 | 0.688 | | 1.112 | 0.268 |
| $X_1$ Percentage ICT Access | 0.152 | 0.013 | 0.834 | 11.767 | 0.000 *** |
| $X_2$ Percentage Non-ICT Access | −0.088 | 0.013 | −0.287 | −6.943 | 0.000 *** |
| $X_3$ Gender | 0.537 | 0.270 | 0.085 | 1.986 | 0.049 ** |
| $X_4$ Educational level | 0.030 | 0.062 | 0.036 | 0.485 | 0.629 |
| $X_5$ Family labor | −0.056 | 0.039 | −0.068 | −1.443 | 0.152 |
| $X_6$ Relative Farming Experience | 0.291 | 0.564 | 0.025 | 0.517 | 0.606 |
| $X_7$ Landholding size | −0.001 | 0.005 | −0.007 | −0.177 | 0.860 |
| $X_8$ Employment type | −0.227 | 0.169 | −0.075 | −1.346 | 0.181 |
| $X_9$ Off-farm income | −0.00005 | 0.000 | −0.060 | −1.031 | 0.305 |
| $R^2$ | 0.821 | | | | |
| Adjusted $R^2$ | 0.808 | | | | |
| F statistic | 62.701 | | | | |
| N | 133 | | | | |

Dependent Variable: Marketing Information (Mean). *** and ** denote significance at 1% and 5% levels respectively.

## 4. Conclusions

This study examined famers' access to production and marketing information in the Semi-Arid Region of Rajasthan State in India. Primary data was collected using a multistage sampling procedure

and questionnaire administration from 133 farmers in Jalore, Pali and Sirohi districts in the Rajasthan State. The study examined the differences in farmers' access to different types of marketing and production information using the Analysis of Variance test. The study also examined the differences in access to each type production and marketing information between two categories of farmers (i.e., farmers who sought information from ICT sources and those that sought information from Non-ICT sources) using the student's *t*-test. Finally, the study examined the overall effects of the percentage of information sought from ICT sources, and the percentage of information sought from Non-ICT sources on famers' access to production and marketing information. The results of the Analysis of Variance test regarding the farmers' access to different types of production and marketing information revealed that the user type (i.e., ICT versus Non-ICT users) significantly explains the differences in farmers' access to the different types of marketing and production information. These results are consistent with the empirical results of the student's *t*-test that farmers' access to different types of production and marketing information from ICT sources is significantly higher than from Non-ICT sources. Consistently, the empirical results of the multiple regressions revealed that the percentage of production and marketing information obtained from ICT sources had a positive significant influence on the farmers' level of access to marketing and production information. On the contrary, the percentage of marketing and production information obtained from Non-ICT sources had a negative significant influence on the farmers' level of access to marketing and production information. These results suggest that ICT sources of marketing and production information play a crucial role in the farmers' access to this information for their business operations and that the male farmers have more access to production and marketing information than the female farmers.

The policy recommendation from this study is as follows. It is imperative that the government through its agencies in the agricultural sector train the farmers (especially the female farmers) to seek information from ICT sources to obtain adequate information for their production and marketing operations. In this respect, the crucial role of the Extension Department of the Ministry of Agriculture in the training programs is a step in the right direction. Also, the literacy level of the illiterate farmers can be improved through informal education and training to enable them to understand the means of accessing production and marketing information from ICT sources. The combined crucial role of the Ministry of Agriculture and adult education department of the Ministry of Education would be very important in raising the literacy level of the farmers. Finally, public-private partnership in the provision of ICT infrastructure to make adequate production and marketing information available to the famers through ICT sources is also very important.

The key limitation of this study is its coverage, i.e., the data is only from the sample of farmers in three districts in Rajasthan State in India. Therefore, extending the research area to other regions within India to provide more general conclusions is an excellent opportunity for future research.

**Author Contributions:** Conceptualization, I.S.P. and P.S.; Data curation, I.S.P., P.S., J.K.M.K. and K.R.S.; Formal analysis, I.S.P., P.S., J.K.M.K.; Funding acquisition, P.S., J.K.M.K. and K.R.S.; Investigation, P.S., J.K.M.K. and K.R.S.; Project administration, P.S. and J.K.M.K.; Resources, I.S.P. and P.S.; Supervision, P.S. and K.R.S.; Validation, I.S.P., P.S., J.K.M.K. and K.R.S.; Visualization, P.S. and J.K.M.K.; Writing—original draft, I.S.P.; Writing—review and editing, I.S.P., P.S., J.K.M.K. and K.R.S.

**Acknowledgments:** The authors are grateful to the anonymous reviewers for their fruitful suggestions. Encouraging support from Narendra Singh Rathore, Deputy Director General (Education), Indian Council of Agricultural Research (ICAR), India is sincerely acknowledged. Support from the Farm Science Centers of Sumerpur and Sirohi region in collecting data is thankfully acknowledged.

**Conflicts of Interest:** The authors declare no conflict of interest.

## Appendix A

**Table A1.** Descriptive statistics of farmers' access to different types of production information.

|  | **Mean** | **Std. Deviation** |
|---|---|---|
| 1. Input: price and availability | 2.80 | 1.481 |
| 2. Seed: best variety, use (kg/acre), treatment | 2.89 | 1.455 |
| 3. Weather: temperature and rainfall | 3.19 | 1.513 |
| 4. Soil: preparation, testing | 2.83 | 1.443 |
| 5. Sowing: time, method | 2.77 | 1.492 |
| 6. Fertilizer: type, dosage, time, method | 2.86 | 1.478 |
| 7. Irrigation: time, number, method | 3.14 | 1.613 |
| 8. Pesticide/herbicide type, dosage, time, method | 2.77 | 1.433 |
| 9. Harvesting: time, method | 3.12 | 1.518 |

**Table A2.** Descriptive statistics of farmers' access to different types of marketing information.

|  | **Mean** | **Std. Deviation** |
|---|---|---|
| 1. Storage: cost, method | 2.68 | 1.514 |
| 2. Packing: cost, method | 2.74 | 1.565 |
| 3. Transportation: types, sources and costs | 2.83 | 1.478 |
| 4. Market price | 2.90 | 1.532 |
| 5. Farm gate price | 2.59 | 1.451 |
| 6. Market place | 2.98 | 1.532 |
| 7. Market charges | 2.80 | 1.536 |
| 8. Future prices | 2.83 | 1.610 |
| 9. Buyer and Trader contact | 2.58 | 1.447 |

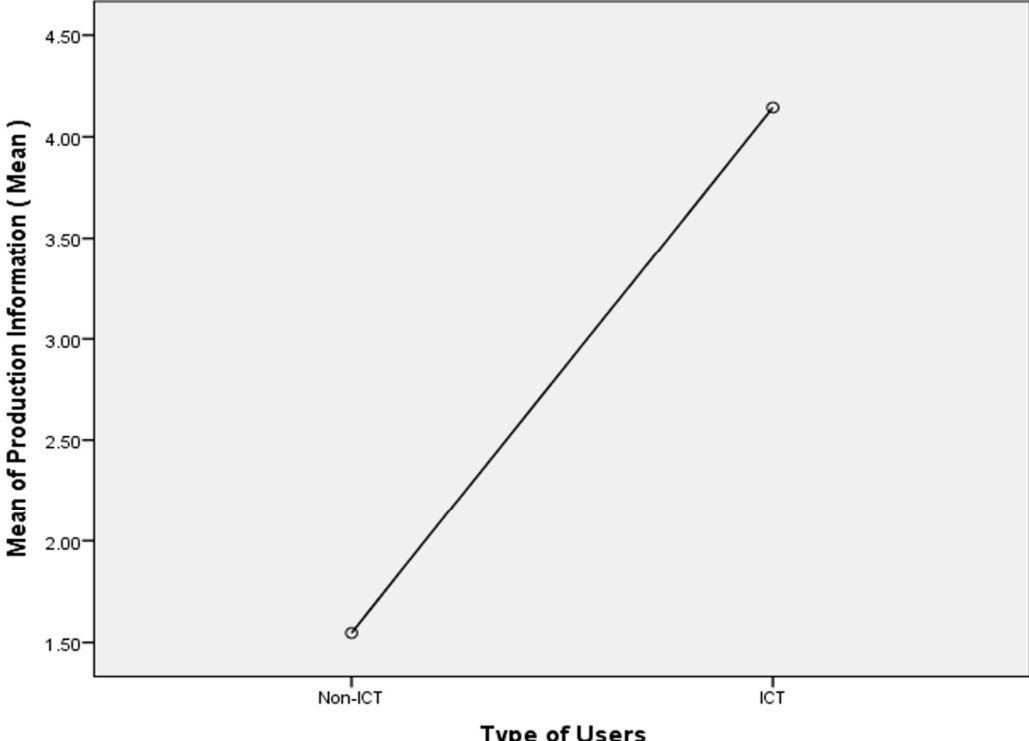

**Figure A1.** Estimated marginal means of level of access of production information (mean) vs. type of users.

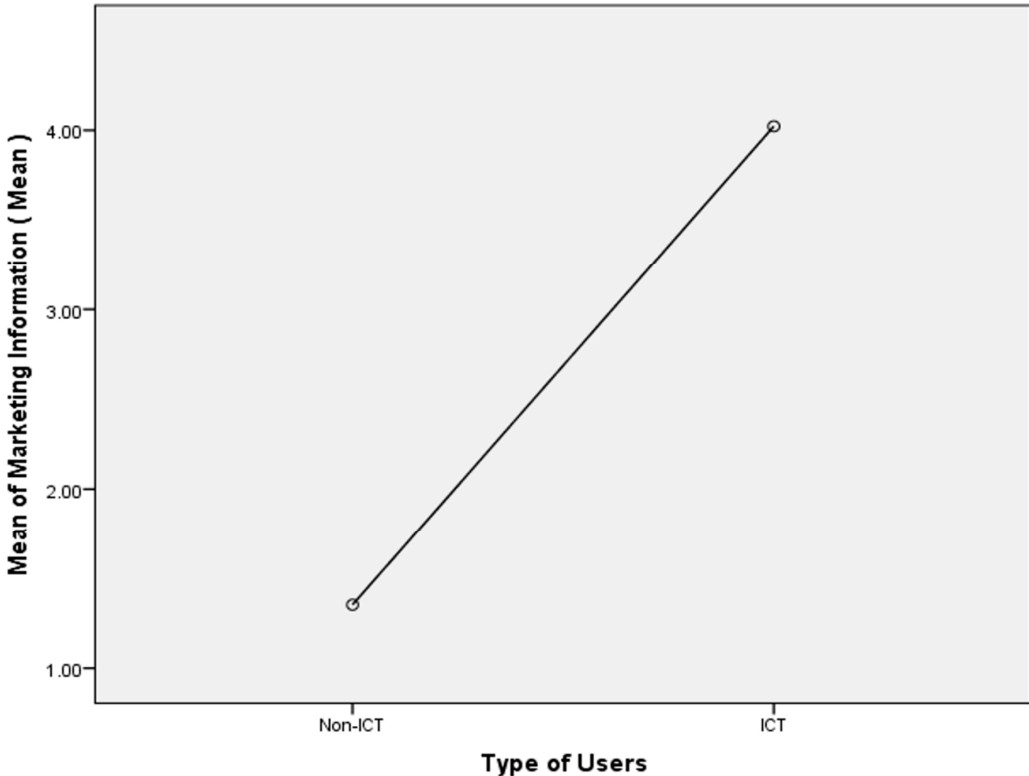

**Figure A2.** Estimated marginal means of level of access of marketing information (mean) vs. type of users.

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
