# Peer review of "Evaluating Farmers’ Access to Agricultural Information: Evidence from Semi-Arid Region of Rajasthan State, India"

_agriculture, doi:10.3390/agriculture9030060_

Round 1
Reviewer 1 Report
Summary: The authors examine whether the availability/use of ICTs improves access to information among rural farmers in Rajasthan. They differentiate between information on agricultural production and information on agricultural marketing in the analysis —and between different categories of information on production and marketing. The authors find that ICTs users make greater use of the different sources of information and thus recommend improving farmers´ skills in the use of ICTs in order to increase their use of information.
While I find that the role of ICTs on farmers´ access to information can be of key importance for policy making, I think the paper in its current version can be improved in several aspects:
Broad comments:
I miss a description of the economic theory behind the article, as well as of the background of the study —area of study and other features which may affect the results of the study and their implications for the analysis. I would suggest the authors providing some information on both.
I think more information about the survey and the data should be provided —more detail on the variables collected in the survey, some information about the survey´s design, whether it is nationally representative or to what extent it is representative, and the source of the relevant auxiliary information which was “acquired from reliable sources” (line 105).
I would suggest the authors using a binary variable instead of X1 and X2 in the regression analysis. I would also suggest considering the possibility of using additional control variables in the regression in order to gain additional insight into the study.
Specific comments:
· I suggest not using so many subsections in Section 2.
· I think the specification of the WAI should be better explained (lines 131-145). Were the “different levels of access” used in its calculation collected in the survey? Please indicate whether each category of information used in the analysis is assigned one of these levels of access in the survey. Otherwise provide information on how the variables are computed.
· Is the dependant variable Y used in the regression the WAI? If so, the authors should say so.
· Line 153: do you refer by any chance to household heads?
· Lines 153-171: I would suggest including an additional table which contains the descriptive statistics. It would improve the paper.
· Outline how the “Production Information (mean)” and the “Marketing Information (mean)” are defined (from line 177 on).
· I would also suggest mentioning the fact that the authors are differentiating between information on production and information on marketing —and the categories or items they are considering within each of them— earlier in the paper.
· I would suggest including the fact that the authors are undertaking the analysis on multiple comparisons too in the section on material and methods (the results of the analysis are included in Tables 2 and 5) and giving an explanation for its inclusion in the analysis.
· Figures 1, 2 and 3: Could you please explain what the equivalent rank is?
· Please number the equations.
Author Response
Editor | |
Comment | Response |
Reviewer 1 | |
Summary: The authors examine whether the availability/use of ICTs improves access to information among rural farmers in Rajasthan. They differentiate between information on agricultural production and information on agricultural marketing in the analysis —and between different categories of information on production and marketing. The authors find that ICTs users make greater use of the different sources of information and thus recommend improving farmers´ skills in the use of ICTs in order to increase their use of information.
While I find that the role of ICTs on farmers´ access to information can be of key importance for policy making, I think the paper in its current version can be improved in several aspects: | Thank you very much for your encouraging comments. |
Broad comments: I miss a description of the economic theory behind the article, as well as of the background of the study —area of study and other features which may affect the results of the study and their implications for the analysis. I would suggest the authors providing some information on both.
I think more information about the survey and the data should be provided —more detail on the variables collected in the survey, some information about the survey´s design, whether it is nationally representative or to what extent it is representative, and the source of the relevant auxiliary information which was “acquired from reliable sources” (line 105).
I would suggest the authors using a binary variable instead of X1 and X2 in the regression analysis.
I would also suggest considering the possibility of using additional control variables in the regression in order to gain additional insight into the study. | Thank you very much for your critical comment. Please we have now specified the economic rationale behind the article as follows:
The economic rationale for the farmers’ access to information is to enable them to the manage risks and uncertainties regarding production and marketing of their produce. The better the farmers manage these risks and uncertainties the more profitable their businesses become. ICT facilitates awareness and access to market information among the farmers [4]. There are more than 200 ICT development agencies in different stages of implementation in India e.g., Bhoomi, Drishtee. These agencies provide information relating to for instance, climate reports, and marketing information e.g., Krishi Vigyan Kendras/Farm Science Centers at Ahmednagar, Baramati) [11]. (Please see page 3 lines 82 – 88).
We have also provided further information on the area of study. (Please page 4-7).
Please, we have now provided detailed information about the survey including the sampling procedure, population and sample size. (Please see page 8).
The survey was carried out in three districts in Rajasthan State in India. Technically, the data and results obtained in this study may not be fully representative of the population of India. Therefore, we have specified this as the key limitation of the study and provided suggestions for future research in the conclusions (see Section 4, page 18 – 19).
Also, we have now referenced all sources of reliable information that we have obtained for this study. Thank you very much for this comment. Please the data obtained for this study is not binary. The five-point Likert questionnaire was used to obtain the data. Thereafter, the values were averaged using the weighted average as specified on page 8.
In this study, the additional explanatory variables were not used for the regression analysis. Previous studies have examined these relationships as we have noted on page 3 lines 90 – 95. The crux of this study is to examine whether information from ICT sources influence the farmers’ overall access to production and marketing information for their businesses. |
Specific comments | Response |
Point 1: I suggest not using so many subsections in Section 2. | Thank you very much. Please we have now reduced the number of subsections in Section 2 and other parts of the manuscript as suggested. |
Point 2: I think the specification of the WAI should be better explained (lines 131-145). Were the “different levels of access” used in its calculation collected in the survey? Please indicate whether each category of information used in the analysis is assigned one of these levels of access in the survey. Otherwise provide information on how the variables are computed. | Thank you very much. Yes, the different levels of access used in its calculation were collected during the survey.
The WAI is used calculate the level of access of individual production and marketing Information separately. Thus, access to production and marketing information were evaluated based on the data collected by using the weighted average index (WAI) specification on page 8 line 228 – 235). |
Point 3: Is the dependant variable Y used in the regression the WAI? If so, the authors should say so. | Thank you very much. Please No.
The WAI was calculated for each type of production and marketing Information.
Thereafter, the averages of all types of production information were used to derive the production mean for the first multiple regression analysis.
Similarly, the averages of all types of marketing information were used to derive the marketing information mean for second multiple regression analysis.
Please we have now indicated this as a footnote 3 on page 9. |
Point 4: Line 153: do you refer by any chance to household heads? | Thank you very much. Yes, household head. These farmers are main source of income and take major decisions for their households. |
Point 5: Lines 153-171: I would suggest including an additional table which contains the descriptive statistics. It would improve the paper. | Thank you very much. The table of descriptive statistics has now been added to provide further insight. (Please see section 3.1, Table 2, page 9 – 11). |
Point 6: Outline how the “Production Information (mean)” and the “Marketing Information (mean)” are defined (from line 177 on). | Thank you very much.
Production mean is the average of all types of production information accessed.
Marketing mean is the average of all types of marketing information accessed. Please refer to our response to point 3 above. |
Point 7: I would also suggest mentioning the fact that the authors are differentiating between information on production and information on marketing —and the categories or items they are considering within each of them— earlier in the paper. | Thank you very much. Please This information is has now been added to the presentation of the Conceptual Framework of Research.
(Please see page 4). |
Point 8: I would suggest including the fact that the authors are undertaking the analysis on multiple comparisons too in the section on material and methods (the results of the analysis are included in Tables 2 and 5) and giving an explanation for its inclusion in the analysis. | Thank you very much. Please the methods of analysis have now been properly presented in the revised manuscript as suggested.
(Please see page 7 – 9). |
Point 9: Figures 1, 2 and 3: Could you please explain what the equivalent rank is? | Thank you very much. The equivalent rank is the percentage equivalent of ICT source accessed out of Total (ICT+ Non-ICT). |
Point 10: Please number the equations. | Thank you very much. Please we now sequentially numbered all the equations in the manuscript. |
Reviewer 2 Report
Below, I provide some comments that can improve the quality of the manuscript:
1) The authors should first provide a theoretical background/framework of the study than can help them identify the objectives of the paper. The study of the socioeconomic characteristics of the respondents cannot be a scientific objective; this is a tool for the analysis. The authors should outline the role of farm training/ farm education on farm economic performance/ labour productivity (there are a lot of references in the international literature, e.g., Exploring the labour productivity of agricultural systems across European regions: A multilevel approach) and then explore the role of ICTs on achieving this.
2) The authors should enhance their multiple regression model. All the variables derived from the socioeconomic survey (e.g., age, education, perceptions etc.) should be included as explanatory variables (control variables) in the model to test their impact. Also instead of using X1 and X2 for access/not access to ICT, the authors could think of using a binary variable (1: access, 0: otherwise) avoiding thus multicollinearity issues
3) The authors should provide more information about the area of the study and mainly the agricultural sector: contribution of agriculture to total GDP and employment, prospects, dynamics.
4) The authors should provide more details about the design of the survey. What is the farm population size of the case study and what the 133 respondents represent; synthesis of livestock breeders and farmers etc.
5) The article is missing policy recommendations.
Author Response
Reviewer 2 | |
Point 1: The authors should first provide a theoretical background/framework of the study than can help them identify the objectives of the paper. The study of the socioeconomic characteristics of the respondents cannot be a scientific objective; this is a tool for the analysis. The authors should outline the role of farm training/ farm education on farm economic performance/ labour productivity (there are a lot of references in the international literature, e.g., Exploring the labour productivity of agricultural systems across European regions: A multilevel approach) and then explore the role of ICTs on achieving this. | Thank you very much. The theoretical background and conceptual framework of the study have now been provided. (Please see page 3 and 4).
The socioeconomic characteristics of respondents have been removed from the list of objectives as suggested. (Please see page 4).
Also, an exposition of the influence of education and other socio-economic factors as well as farm level factors on ICT access have been provided in the revised manuscript.(Please see page 3). |
Point 2: The authors should enhance their multiple regression model. All the variables derived from the socioeconomic survey (e.g., age, education, perceptions etc.) should be included as explanatory variables (control variables) in the model to test their impact. Also instead of using X1 and X2 for access/not access to ICT, the authors could think of using a binary variable (1: access, 0: otherwise) avoiding thus multicollinearity issues | Thank you very much for this comment. Please the data obtained for this study is not binary. The five-point Likert questionnaire was used to obtain the data. Thereafter, the values were averaged using the weighted average as specified on page 8.
In this study, the additional explanatory variables were not used for the regression analysis. Previous studies have examined these relationships as we have noted on page 3 lines 90 – 95. The crux of this study is to examine whether information from ICT sources influence the farmers’ overall access to production and marketing information for their businesses. |
Point 3: The authors should provide more information about the area of the study and mainly the agricultural sector: contribution of agriculture to total GDP and employment, prospects, dynamics. | Thank you very much. Please we have now provided more information about area of study. (Please see section 3.1). |
Point 4: The authors should provide more details about the design of the survey. What is the farm population size of the case study and what the 133 respondents represent; synthesis of livestock breeders and farmers etc. | Thank you very much. Please, further information on the design of survey, farm population size and information about 133 respondents have now been provided. (Please see page 8).
|
Point 5: The article is missing policy recommendations. | Thank you very much. Please, the policy recommendation emanating from the conclusions of the study has now been provided. (Please see section 4, page 19-20). |
Round 2
Reviewer 1 Report
Even though the paper has improved, I still have a few concerns:
· I would suggest the authors combining both maps into one (Figures 3 and 4).
· In the description of variables section: Why are equations 1 and 2 divided by 2? Besides, I am not sure I would consider information obtained through own knowledge and experience as “information”.
· I would suggest including Table 2 in the “sampling and data” section or in a new different section (descriptive statistics are not results). I think it could be a good idea to include statistics for both users and non-users in two different columns. It would give additional insights into the study.
· In the original paper three groups of ICT users were considered (ICT users/non-ICT users/moderate users). In the reviewed manuscript the authors consider two only. I assume moderate users have been included in the “ICT users” group. The Games-Howell post-hoc test does not longer make sense. It is used to confirm which means (between the different groups) are significantly different from each other. The ANOVA already confirms there are differences between the two groups analysed. If the authors still want to include the Games-Howell test, I would suggest removing the “moderate” category.
· As for the regression, I would still suggest
using some additional explanatory variables from the survey (if it is possible).
Author Response
Reviewer 1 | |
Even though the paper has improved, I still have a few concerns. | Thank you very much. |
I would suggest the authors combining both maps into one (Figures 3 and 4). | Thank you very much. Please we have now combined the two figures into one as suggested (Please see Figure 3). |
In the description of variables section: Why are equations 1 and 2 divided by 2? Besides, I am not sure I would consider information obtained through own knowledge and experience as “information”.
| Thank you very much for this comment. The reason is as follows: We have used equal number of ICT and Non- ICT sources. The % ICT source from which the information was accessed is calculated based on Total ICT sources. Similarly, the % Non-ICT source from which the information was accessed is calculated based on Total Non-ICT sources. Therefore, to calculate % ICT access out of Total ICT + Non- ICT access, it was divided by 2 as specified in equations (1) and (2). Please we have now included this explanation as a footnote in the manuscript (i.e. footnote 3).
Also, we absolutely agree with you that information obtained through own knowledge and experience cannot be considered as information. Therefore, we have deleted equation (3) from the manuscript, more so, that this is not part of the discussion in the manuscript. |
I would suggest including Table 2 in the “sampling and data” section or in a new different section (descriptive statistics are not results). I think it could be a good idea to include statistics for both users and non-users in two different columns. It would give additional insights into the study. | Thank you very much for this suggestion. Please we have now included the socio-economic characteristics of the farmers as part of section 2 of the manuscript as suggested and categorized them into farmers who solicited information from ICT sources and Non-ICT sources. |
In the original paper three groups of ICT users were considered (ICT users/non-ICT users/moderate users). In the reviewed manuscript the authors consider two only. I assume moderate users have been included in the “ICT users” group. The Games-Howell post-hoc test does not longer make sense. It is used to confirm which means (between the different groups) are significantly different from each other. The ANOVA already confirms there are differences between the two groups analysed. If the authors still want to include the Games-Howell test, I would suggest removing the “moderate” category.
| Thank you very much for this comment.
Please we have added the moderate ICT users to the “ICT users” group, and have removed the moderate category from the discussion. The moderate users were only 3 in our sample; therefore, we thought it would be proper to add them to the ICT users group. Consequently the presentation in this revised manuscript is based on ICT users and Non-ICT users. Consequently, we have now replaced the Games-Howell post-hoc test with the Student’s t-test of the differences in access to different types of information between ICT users and Non-ICT users (Please see Table 4 and 7). |
As for the regression, I would still suggest using some additional explanatory variables from the survey (if it is possible). | Thank you very much. Please we have now included socio-economic and some farm-level variables in the regression (Table 5 and 8). The gender variable is significant in explaining farmers’ access to production and marketing information, and the corresponding implication has been provided. |
Reviewer 2 Report
The authors have addressed most of the comments
Author Response
Reviewer 2 | |
The authors have addressed most of the comments | Thank you very much. |